# The PINK1—Parkin mitophagy signalling pathway is not functional in peripheral blood mononuclear cells

**Aaron V. Bradshaw, Philip Campbell**, **Anthony H. V. Schapira, Huw R. Morris, Jan-Willem Taanman** *

Department of Clinical and Movement Neurosciences, Queen Square Institute of Neurology, University College London, London, United Kingdom

* j.taanman@ucl.ac.uk

## Abstract

Mutations in the *PINK1* and *PRKN* genes are the most common cause of early-onset familial Parkinson disease. These genes code for the PINK1 and Parkin proteins, respectively, which are involved in the degradation of dysfunctional mitochondria through mitophagy. An early step in PINK1 –Parkin mediated mitophagy is the ubiquitination of the mitofusin proteins MFN1 and -2. The ubiquitination of MFN1 and -2 in patient samples may therefore serve as a biomarker to determine the functional effects of *PINK1* and *PRKN* mutations, and to screen idiopathic patients for potential mitophagy defects. We aimed to characterise the expression of the PINK1 –Parkin mitophagy machinery in peripheral blood mononuclear cells (PBMCs) and assess if these cells could serve as a platform to evaluate mitophagy via analysis of MFN1 and -2 ubiquitination. Mitophagy was induced through mitochondrial depolarisation by treatment with the protonophore CCCP and ubiquitinated MFN proteins were analysed by western blotting. In addition, *PINK1* and *PRKN* mRNA and protein expression levels were characterised with reverse transcriptase quantitative PCR and western blotting, respectively. Whilst CCCP treatment led to MFN ubiquitination in primary fibroblasts, SH-SY5Y neuroblastoma cells and Jurkat leukaemic cells, treatment of PBMCs did not induce ubiquitination of MFN. *PRKN* mRNA and protein was readily detectable in PBMCs at comparable levels to those observed in Jurkat and fibroblast cells. In contrast, PINK1 protein was undetectable and *PINK1* mRNA levels were remarkably low in control PBMCs. Our findings suggest that the PINK1 –Parkin mitophagy signalling pathway is not functional in PBMCs. Therefore, PBMCs are not a suitable biosample for analysis of mitophagy function in Parkinson disease patients.

## Introduction

Parkinson disease (PD) has a heterogeneous clinical presentation. An important goal in PD research is to identify if and how this heterogeneity is related to biochemical and cell biological differences, which inform the pathophysiology of the disease [1]. The identification of such

**Data Availability Statement:** All relevant data are within the manuscript and its Supporting Information files.

**Funding:** JWT, 15271, Michael J. Fox Foundation for Parkinson's Research, https://www.michaeljfox.org/, NO JWT, 42, Royal Free Charity, https://royalfreecharity.org/, NO AHVS, 668738, European Union Horizon 2020 Research and Innovation programme https://ec.europa.eu/programmes/horizon2020/en/home, NO AHVS, MR/M006646/1, Medical Research Council, https://mrc.ukri.org/, NO The funders had no role in study design, data collection and analysis, decision to publish, or preparation of the manuscript.

**Competing interests:** I have read the journal's policy and the authors of this manuscript have the following competing interests: JWT received funding from Michael J. Fox Foundation for Parkinson's Research and the Royal Free Charity. AHVS received funding from the European Union and the Medical Research Council, and is a Consultant for Sanofi Aventis. HM is a Consultant for AlzProtect, Accorda, Bristol-Myers-Squibb, E-scape and the Wellcome Trust, and has received lecture fees from GE Healthcare, GSK, UCB Pharma, and the Wellcome Trust. This does not alter our adherence to PLOS ONE policies on sharing data and materials.

differences is important for three interrelated reasons: (1) the elucidation of mechanistic targets for drug development, (2) the selection of the most appropriate patient cohorts for clinical trials, and (3) to provide functional readouts allowing the identification of target engagement and changes in cellular and biochemical phenotypes.

Although most cases of PD are idiopathic, there are some rare early-onset familial forms, which may provide insights into potential biochemical subtypes that exist within the wider, idiopathic PD population. The most common forms of early-onset familial PD are caused mutations in the *PINK1* and *PRKN* genes, which encode PTEN-induced putative kinase 1 (PINK1) [2] and Parkin, respectively [3]. Both proteins act in the same quality control pathway to sense damaged mitochondria and target them for degradation through a specialised form of macro-autophagy, also known as mitophagy. PINK1 is a mitochondrial kinase imported into the mitochondria via the preprotein translocase complexes where it is constitutively degraded by the presenilin-associated rhomboid-like protein (PARL) [4]. Loss of mitochondrial membrane potential disrupts the mitochondrial import and degradation of PINK1, which subsequently accumulates on the outer mitochondrial membrane (OMM) [5]. Here, PINK1 undergoes autophosphorylation [6] and phosphorylates Parkin at serine 65, promoting its mitochondrial translocation and stabilisation [7]. Phosphorylation of ubiquitin at serine 65 by PINK1 further promotes the full activation of Parkin [8, 9]. Parkin, an E3 ubiquitin ligase ubiquitinates multiple OMM proteins, including mitofusin 1 (MFN1) and mitofusin 2 (MFN2) [10]. Recruitment of autophagy adaptors by ubiquitin chains conjugated to OMM proteins [11] leads to the engulfment of impaired mitochondria by autophagosomes, which fuse with lysosomes causing the eventual degradation of defective mitochondria.

The clinical, molecular and functional genetics of *PINK1* and *PRKN* are complex. Disease causing mutations can include missense mutations, deletions and copy number variations, which can be inherited in homozygous and compound heterozygous patterns [2, 3, 12, 13]. Pathogenic *PINK1* and *PRKN* mutations are generally associated with a loss of function of the respective proteins. Furthermore, recent findings demonstrate that heterozygously inherited *PINK1* mutations can confer increased PD risk, an effect that may be mediated at the molecular level by a dominant negative mechanism during PINK1 dimerisation [14, 15]. However, the effects of these mutations on Parkin and PINK1 function, and on downstream mitophagy, have not been fully characterised. For instance, there are over 200 *PRKN* variants, some of which are pathological and which have differing effects on Parkin function and mitophagy [16]. Moreover, mitophagy is impaired in skin fibroblasts from PD patients with no known genetic cause [17, 18]. These findings suggest that defects in mitophagy may contribute to idiopathic PD in some patients [19].

Although the role of mitophagy in the pathogenesis of PD is debated, functional readouts of the PINK1–Parkin mitophagy signalling pathway may provide patient stratification, even if they are epiphenomenal to disease causing processes. Such readouts will also allow the functional characterisation of novel *PINK1* and *PRKN* variants and those of unknown significance. One such functional readout is the ubiquitination of the MFN proteins following mitochondrial depolarisation, a lack of which distinguishes *PINK1* and *PRKN* mutant fibroblasts from those isolated from healthy controls [20]. In this study we aimed to translate this readout to peripheral blood mononuclear cells (PBMCs), which represent a minimally invasive source of biological tissue for biomarker analysis. We have also characterised the expression of the PINK1–Parkin signalling pathway in different cell types and demonstrate that the pathway is not functional in PBMCs. This work thus adds an important contribution to knowledge on the mitochondrial biology of PBMCs and their utility in PD research.

## Results

### Treatment of fibroblasts with CCCP induces MFN ubiquitination in a PINK/Parkin dependent manner

We began by characterising the effect of mitochondrial depolarisation upon the ubiquitination of the OMM proteins MFN1 and MFN2 in primary fibroblast cultures. Fibroblasts from 2 healthy controls were treated with 20 μM carbonyl cyanide *m*-chlorophenyl hydrazone (CCCP) for 2 hours to depolarise the mitochondria followed by western blot analysis. In untreated cells, MFN1 was detected as a double band, with the lower and upper bands migrating at 75 and 78 kDa, respectively, and MFN2 was detected as a single band with an apparent $M_r$ of 75 kDa (Fig 1). CCCP treatment led to the appearance of an extra anti-MFN1 reactive band, which migrated with an apparent $M_r$ that was 9 kDa larger than the lower MFN1 band detected in untreated cells. In addition, an extra anti-MFN2 reactive band that was 9 kDa larger than the MFN2 band in untreated cells was detected (Fig 1). This size change is consistent with mono-ubiquitination and our previous anti-mitofusin immunoprecipitation experiments [10] indicated that the extra MFN1 and MFN2 bands detected post-CCCP treatment are ubiquitin positive. Next, we performed the same treatment and analysis on fibroblast cultures from two early-onset PD patients, one with a homozygous nonsense mutation in *PINK1* and one with a homozygous deletion in *PRKN*. Consistent with previous reports [20], CCCP treatment failed to induce ubiquitination of either MFN1 or MFN2 in fibroblasts from either patient (Fig 1). These results demonstrate that CCCP-induced ubiquitination of MFN1 and MFN2 in fibroblasts depends upon the presence of functional PINK1 and Parkin proteins.

### Treatment of Jurkat cells with CCCP induces MFN ubiquitination

We next asked whether CCCP-induced MFN ubiquitination was a shared feature across different cell types. We first opted to analyse the immortalised T-lymphocyte cell line Jurkat as a surrogate model for primary leucocyte cells. Similar to that observed in control fibroblasts, CCCP treatment produced extra anti-MFN1 and anti-MFN2-2 reactive bands, whose migration was consistent with mono-ubiquitination of the respective proteins (Fig 2A). In addition, we observed faint anti-MFN1 and anti-MFN2-2 reactive bands suggestive for polyubiquitination. Time course analysis demonstrated that CCCP-induced ubiquitination of MFN1 and MFN2 occurred as early as 30 minutes post-treatment and persisted up to 24 hours post-treatment (Fig 2B). The maximum ubiquitinated MFN signal was detected at 2 hours post-treatment, for both MFN1 and MFN2 (S1 Fig). Thus, Jurkat cells respond in the same way as fibroblasts to CCCP treatment in terms of MFN ubiquitination.

### Treatment of PBMCs from healthy controls with CCCP does not induce MFN ubiquitination

Following these findings, we aimed to compare the CCCP-induced MFN ubiquitination response across 4 different cell types: the immortalised neuroblastoma SH-SY5Y cell line, the immortalised Jurkat T-lymphocyte cell line, primary cultured fibroblasts and PBMCs derived from healthy controls. Similarly to the results obtained with fibroblasts and Jurkat cells, and consistent with our previous report [10], treatment of SH-SY5Y cells with CCCP produced an anti-MFN1 and anti-MFN2 reactive profile consistent with ubiquitination (Fig 3). As demonstrated in Figs 1 and 2, anti-MFN1 and anti-MFN2 reactive bands consistent with mono-ubiquitination were also detected in CCCP-treated Jurkat cells and fibroblasts; however, we did not detect anti-MFN1 and anti-MFN2 reactive bands consistent with MFN ubiquitination in CCCP-treated PBMC samples (Fig 3). Long exposure images also did not reveal any signal

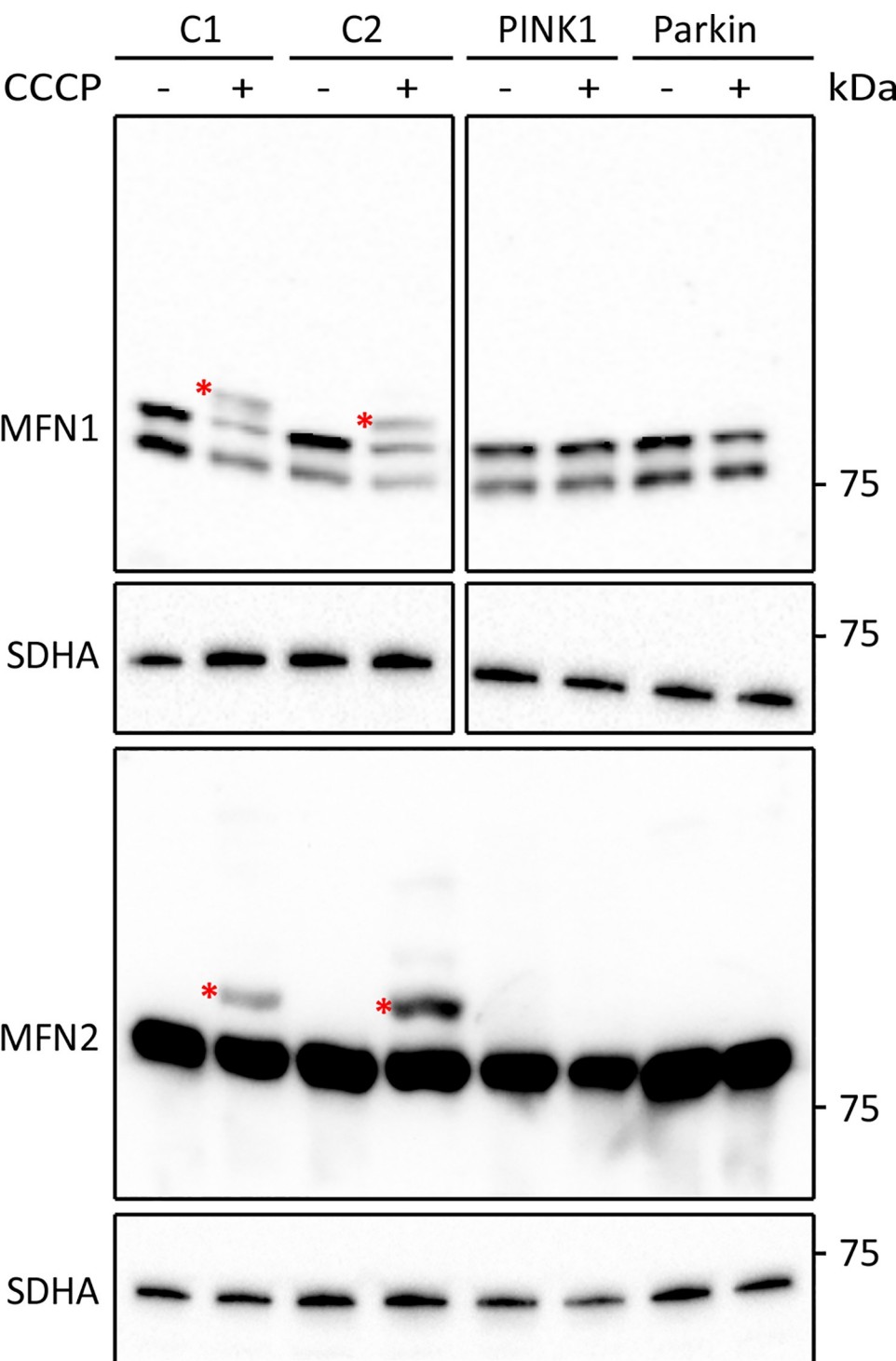

**Fig 1. Analysis of CCCP-induced MFN ubiquitination in cultured fibroblast cells.** Western blot analysis of fibroblasts from healthy controls (C1 and C2) and two early-onset PD patients with mutations in *PINK1* or *PRKN*. Cells were left untreated or were treated for 2 hours with 20 μM CCCP. Blots were developed with antibodies against MFN1 and MFN2. Antibodies against SDHA were used as loading control. Asterisks indicate bands representing ubiquitinated MFN proteins.

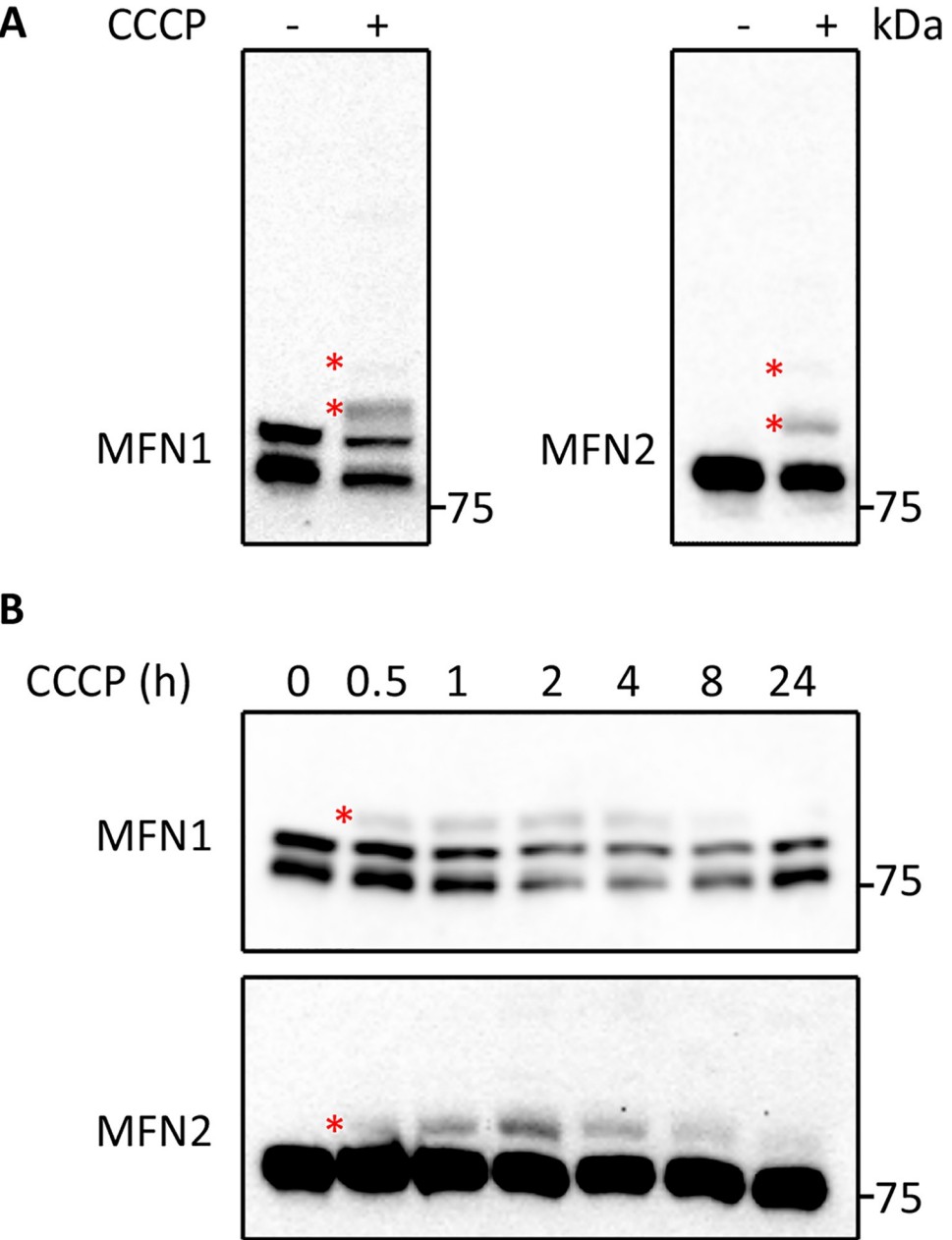

**Fig 2. Analysis of CCCP-induced MFN ubiquitination in Jurkat cells.** (A) Jurkat cells left untreated or treated for 2 hours with 20 μM CCCP were analysed by western blotting with antibodies against MFN1 and MFN2. (B) Western blot analysis of Jurkat cells treated with CCCP for increasing periods of time (0.5–24 hours), developed with antibodies against MFN1 and MFN2. Asterisks indicate bands representing ubiquitinated MFN proteins.

consistent with ubiquitination of either MFN1 or -2. These results demonstrate that under the conditions tested, CCCP treatment does not induce ubiquitination of MFN1 or -2 in PBMCs.

As not all cell types may respond in the same manner to CCCP, we investigated whether the treatment of PMBC samples with 20 μM CCCP is sufficient to promote mitochondrial depolarisation. Jurkat and PBMC samples were loaded with the mitochondrial membrane potential-dependent dye tetramethylrhodamine methylester (TMRM) followed by treatment with CCCP under the same culture conditions used to induce MFN ubiquitination in Jurkat

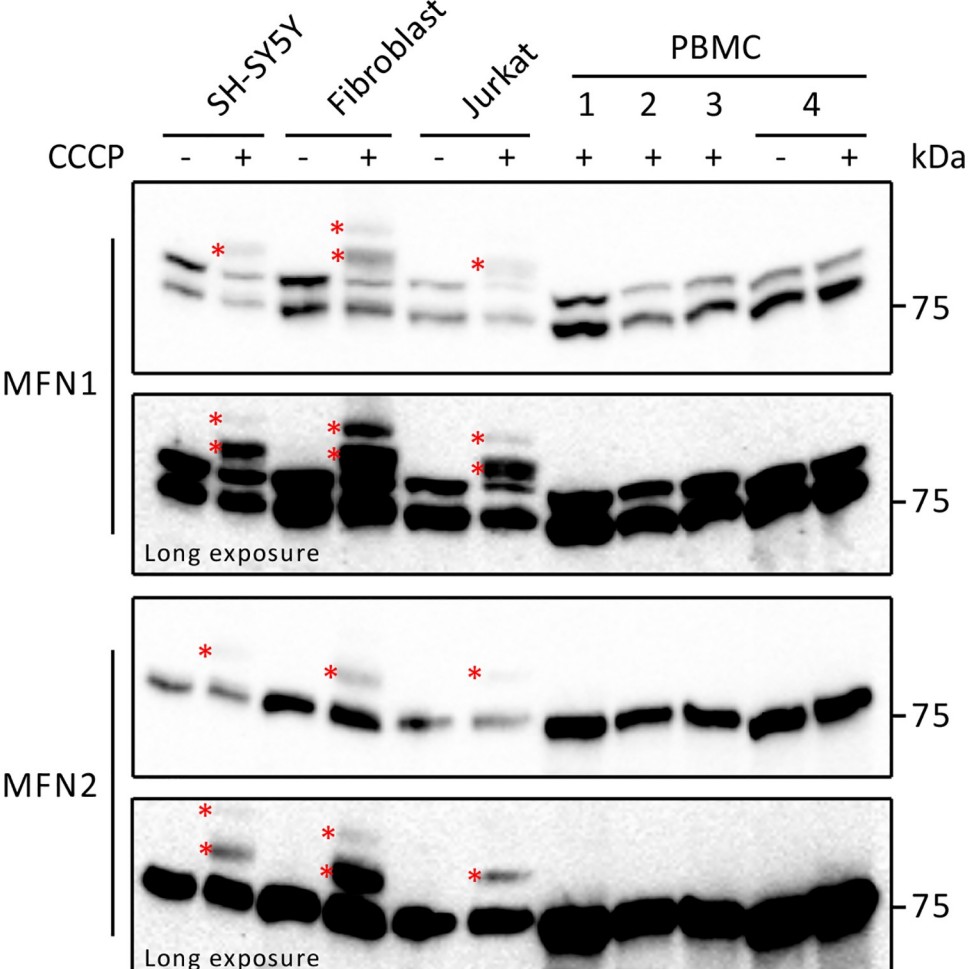

**Fig 3. Analysis of CCCP-induced MFN ubiquitination in different cell types.** SH-SY5Y, fibroblast, Jurkat and PBMC samples were left untreated or were treated with 20 µM CCCP for 2 hours and analysed by western blotting with antibodies against MFN1 and MFN2. Asterisks indicate bands representing ubiquitinated MFN proteins.

cells. Confocal microscopy revealed TMRM-stained, red fluorescent mitochondria in Jurkat and PBMCs under basal condition (Fig 4). Addition of CCCP resulted in a rapid loss of the dye from the mitochondria in both cell types, indicating a rapid dissipation of the

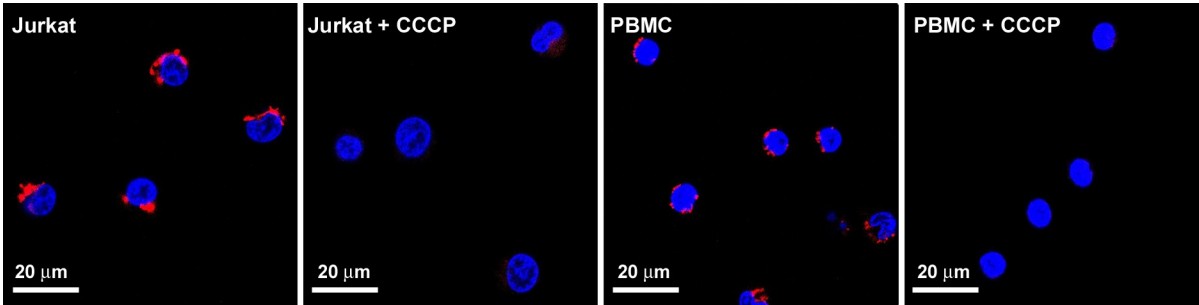

**Fig 4. CCCP induces mitochondrial depolarisation in Jurkat and PBMC samples.** Fluorescent micrographs of Jurkat and PBMC samples loaded with the red fluorescent, mitochondrial membrane potential-dependent dye TMRM before and after addition of 20 µM CCCP (+ CCCP). Nuclei were counterstained fluorescent blue with Hoechst 33343.

mitochondrial membrane potential (Fig 4). This experiment confirms that, under the conditions tested, CCCP treatment induces mitochondrial depolarisation in PBMC samples.

## Parkin is expressed in PBMCs from healthy controls

In order to shed further light on lack of CCCP-induced MFN ubiquitination in PBMCs, we next determined the relative expression of PINK1 and Parkin. We began by assessing the relative abundance of *PRKN* transcripts in Jurkat, fibroblast, SH-SY5Y and PBMC samples. RNA was isolated from the cells and reverse transcribed to create cDNA libraries from which *PRKN* transcripts were amplified by real-time quantitative PCR (qPCR). *PRKN* mRNA was readily amplified from all cell types, and when normalised, fibroblasts (2 independent cultures) and SH-SY5Y cells exhibited 11.8, 10.2 and 12.6 times the *PRKN* levels detected in Jurkat cells (Fig 5A). *PRKN* transcripts were also amplified from 5 independent control PBMC cultures. Levels in these cultures were similar to those detected in Jurkat cells (Fig 5A).

By western blot analysis, Parkin protein was readily detected in 3 independent control fibroblast cultures as well as in Jurkat cells and SH-SY5Y cells as a single band migrating at ~48 kDa (Fig 5B). SH-SY5Y cells demonstrated the highest Parkin signal (~23 fold more than Jurkat and fibroblast cells, Fig 4D). A ~48-kDa band was also detected in PBMC samples from 6 healthy controls (Fig 5C). This protein co-migrated with that detected in SH-SY5Y cells and

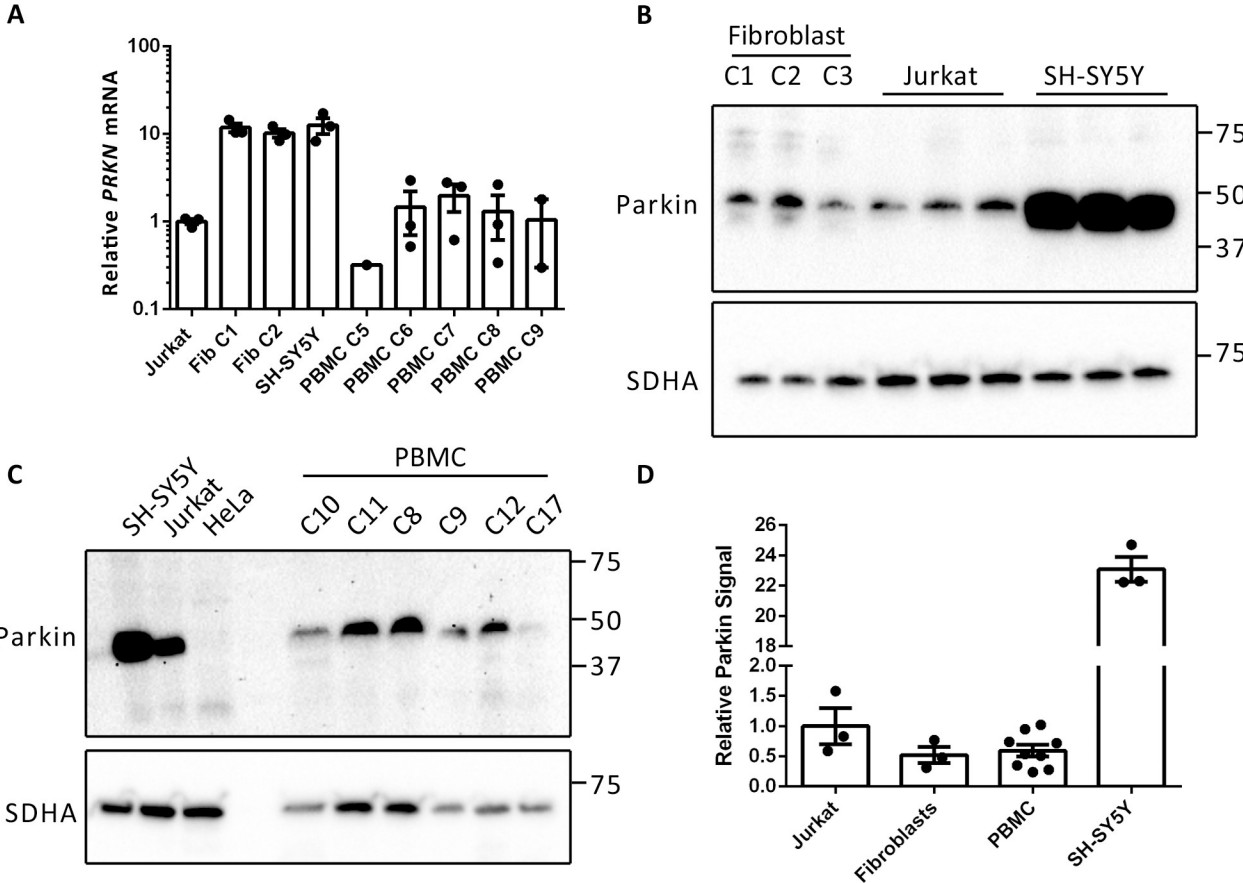

**Fig 5. Analysis of Parkin expression in different cell types.** (A) Reverse transcriptase qPCR analysis of *PRKN* mRNA levels in Jurkat, control fibroblast (Fib), SH-SY5Y and PBMC samples. qPCR was performed on equal amounts of cDNA from each cell type and $C_t$ values were expressed relative to the Jurkat sample. (B, C) Western blot analysis of fibroblast, SH-SY5Y, Jurkat and PBMC cultures with an antibody against Parkin. (D) Quantification of Parkin signal in the different cell types. Signal is expressed relative to SDHA and normalised to that detected in Jurkat cells. Graphs display mean ± SEM.

Jurkat cells, and importantly, was absent from HeLa cells, which have been demonstrated to not express Parkin protein [21]. Therefore, we assume that the ~48-kDa band detected in PBMCs represents Parkin. Quantitatively, the Parkin signal from PBMCs was not significantly different from that detected in either Jurkat or fibroblast cells (Fig 5D). Collectively, these findings demonstrate that Parkin is expressed in PBMCs and, consequently, that the lack of CCCP-induced MFN ubiquitination in these cells is not due to a lack of Parkin expression.

## PINK1 is undetectable in PBMCs from healthy controls

In order to investigate the expression levels of *PINK1* mRNA in PBMCs, RNA was isolated from Jurkat, fibroblast, SH-SY5Y and PBMC samples and followed by reverse transcriptase qPCR. *PINK1* transcript levels in fibroblasts exceeded those detected in Jurkat cells by 33.8 and 32.7 times (for two independent control cultures), whilst levels in SH-SY5Y cells were similar to those in Jurkat cells (Fig 6A). Levels of *PINK1* mRNA in PBMCs from healthy controls were

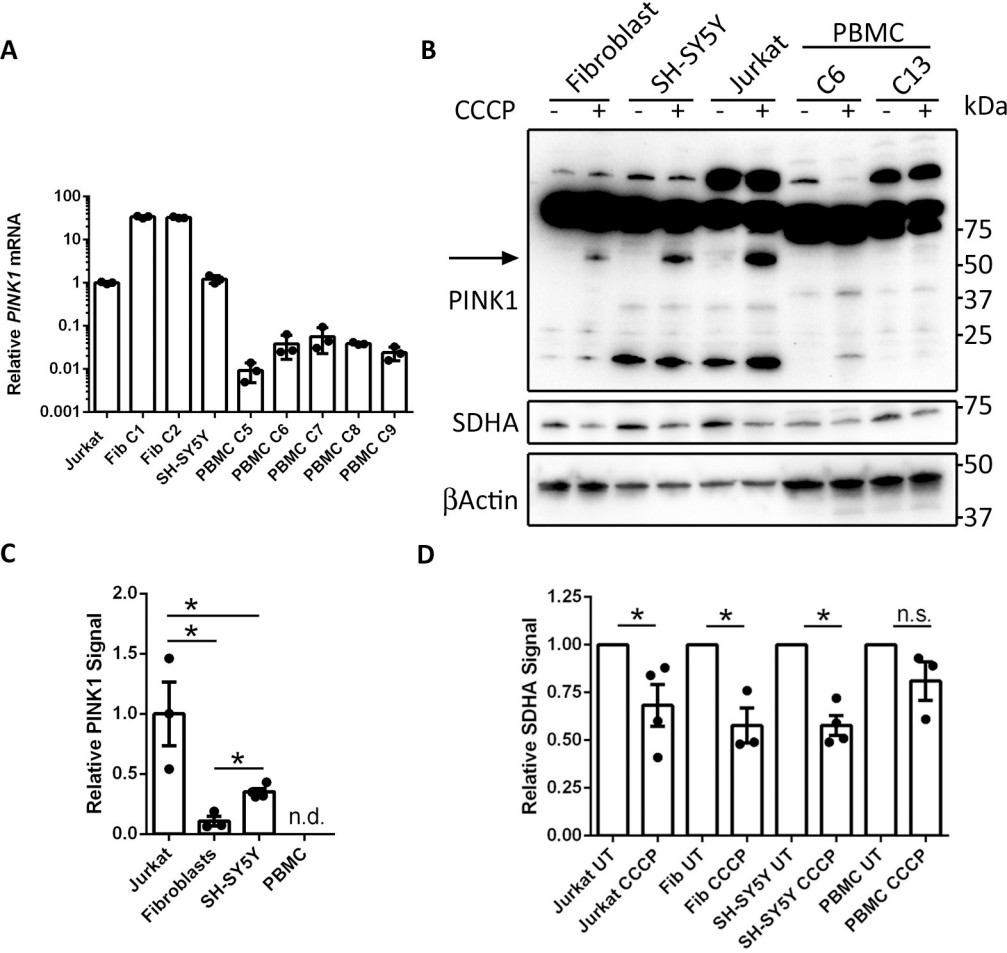

**Fig 6. Analysis of PINK1 expression in different cell types.** (A) Reverse transcriptase qPCR analysis of *PINK1* mRNA levels in Jurkat, control fibroblast (Fib), SH-SY5Y and PBMC samples. qPCR was performed on equal amounts of cDNA from each cell type and $C_t$ values were expressed relative to the Jurkat sample. (B) Western blot analysis of fibroblasts, SH-SY5Y, Jurkat and PBMC cultures left untreated and treated with 20 μM CCCP for 24 hours with an antibody against PINK1 (Novus Biologicals, BC100-494). The band corresponding to PINK1 is indicated by the arrow. (C) Quantification of PINK1 signal from CCCP-treated cells. Signal is expressed relative to β-actin and normalised to levels detected in Jurkat cells. (D) Quantification of SDHA signal from untreated and 24-hour CCCP-treated Jurkat, fibroblast, SH-SY5Y and PBMC samples. Signal is expressed relative to β-actin and normalised to the untreated condition. Graphs display mean ± SEM; * = $P<0.05$, Student's t-test; n.d. = not detected; n.s. = not significant.

20─100 times lower than those detected in Jurkat cells and >3000 times lower than those detected in fibroblast cultures (Fig 6A).

PINK1 protein is constitutively degraded by PARL in the mitochondria and, as a result, undetectable under basal conditions. In order to detect PINK1, the mitochondrial membrane potential must be dissipated, for example with prolonged CCCP treatment. Therefore, control fibroblasts, SH-SY5Y cells, Jurkat cells and PBMCs were treated with 20 μM CCCP for 24 hours, followed by western blot analysis for PINK1. PINK1 antibodies have been reported to give a panoply of non-specific bands when used in western blotting, including at the predicted $M_r$ of PINK1, as determined in knockout cells [22]. Such observations may confound the interpretation of results thus rendering the use of biological controls as critical when studying PINK1 protein. Indeed, even in untreated cells we detected numerous cross-reactive bands in fibroblast, SH-SY5Y and Jurkat samples with a PINK1 antibody from Novus Biologicals (Fig 6B). However, when these cells were treated with CCCP, an additional band appeared at ~60 kDa, the predicted $M_r$ of mature PINK1 (Fig 6B). This band was absent in untreated cells and is, thus, consistent with PINK1. The levels of PINK1 protein were significantly higher in Jurkat cells than either fibroblast or SH-SY5Y cells. Fibroblast cells expressed the lowest levels of PINK1 after CCCP treatment, 10 times lower than detected in Jurkat cells (Fig 6C). However, in PBMCs the pattern of PINK1 cross reactive proteins was unchanged between treated and untreated cells (Fig 6B). Specifically, we did not detect an additional band at 60 kDa when cells were treated with CCCP, thus suggesting that PINK1 protein is not detectable in PBMCs. With an alternative PINK1 antibody from Cell Signaling Technologies, we also failed to detect PINK1 in CCCP-treated PBMCs, despite adequate detection in the other cell types (S2 Fig). The lack of PINK1 detection in PBMCs may explain the absence of CCCP-induced MFN ubiquitination in these cells.

During mitophagy, mitochondrial proteins are degraded in the lysosome. Accordingly, mitophagy can be monitored by assessing the levels of mitochondrial proteins post-mitophagic induction. Indeed, in Jurkat, fibroblast and SH-SY5Y cells, 24-hour CCCP treatment led to significant reductions in subunit A of the mitochondrial succinate dehydrogenase complex (SDHA) (Fig 6D). In PBMCs, conversely, treatment with CCCP did not significantly affect SDHA levels (Fig 6D). These findings suggest differences in CCCP-induced mitophagy between cultured SH-SY5Y, fibroblast and Jurkat cells on the one hand, and PBMCs on the other.

## Parkin is not recruited to depolarised mitochondria in PBMCs

Accumulation of PINK1 in the outer membrane of depolarised mitochondria promotes recruitment of Parkin from the cytosol to the mitochondrial surface [23]. To investigate if Parkin is associated with depolarised mitochondria in PBMCs, vehicle (DMSO)-treated or CCCP-treated Jurkat and PBMC samples were adhered to coverslips, followed by immunocytochemical staining for Parkin and the OMM protein TOMM20 akin earlier reports [24]. Under basal conditions, no co-localisation was observed between Parkin and TOMM20, however, Parkin co-localised with TOMM20 in CCCP-treated Jurkat cells (Fig 7). In contrast, CCCP treatment did not induce co-localisation of Parkin with TOMM20 in PBMCs (Fig 7). These observations suggest that Parkin is not recruited to depolarised mitochondria in PBMCs.

## The PINK1– Parkin mitophagy signalling pathway is not expressed in activated T cells and lymphoblastoid cell lines

PBMCs cultured under normal conditions are non-proliferative, whereas Jurkat cells are robustly proliferating T lymphocyte-like cells. For that reason, we investigated whether

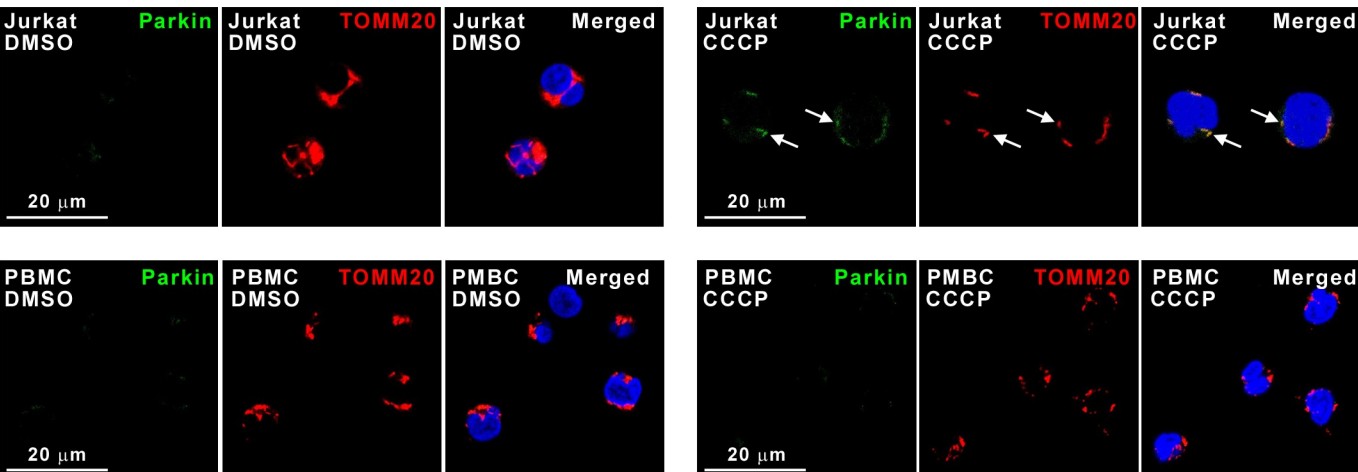

**Fig 7. Parkin does not accumulate on CCCP-induced depolarised PBMC mitochondria.** Fluorescent micrographs of Jurkat and PBMC samples, treated with DMSO (vehicle) or CCCP, followed by immunostaining for Parkin (fluorescent green) and the mitochondrial marker TOMM20 (fluorescent red). Nuclei were counterstained fluorescent blue with DAPI. Arrows indicate Parkin co-localising with mitochondria in Jurkat cells.

inducing the proliferation of T cells in the PBMC samples would shift their response to CCCP treatment. Moreover, we analysed how Epstein-Barr virus (EBV) immortalised lymphoblastoid cell lines (LCLs) from healthy controls responded to CCCP treatment. The T-cell fraction of PBMCs were induced to proliferate by incubation with T-activator CD3/CD28 Dynabeads. This treatment led to the appearance of cell clusters, which were visually similar to cell clusters observed in both Jurkat and LCL cultures (Fig 8A). Despite these morphological similarities, neither activated PBMCs nor LCLs responded to 2-hour CCCP treatment with ubiquitination of MFN1 or MFN2 (Fig 8B). Furthermore, PINK1 protein did not accumulate in either cell type following prolonged CCCP treatment (Fig 8C). *PINK1* mRNA levels in LCLs were also 10–25 times lower than those detected in fibroblasts (S3 Fig). As shown in Fig 5, western blot analysis detected Parkin in Jurkat cells and non-proliferating PBMCs; however, Parkin was undetectable in PBMCs activated with CD23/CD8 Dynabeads and in 4 independent LCLs (Fig 8D). Collectively, these findings demonstrate that whilst Jurkat cells exhibit CCCP-induced PINK1 –Parkin signalling, this pathway is absent from activated PBMCs and LCLs.

## Discussion

In this study we aimed to assess the utility of PBMCs as a platform for studying mitophagy defects in PD patients. Our analysis demonstrated that PBMCs are unsuitable for this purpose as they do not functionally express the PINK1 –Parkin mitophagy signalling pathway. The ubiquitination of MFN proteins following CCCP treatment can be used as a marker of early mitophagy in fibroblasts and distinguishes control fibroblasts from those with *PINK1* or *PRKN* mutations [20]. Our findings demonstrate that this process is not conserved in all cell types. Specifically, PBMCs and lymphocytes immortalised by EBV transduction lack the necessary machinery to transduce loss of mitochondrial membrane potential into MFN ubiquitination.

In agreement with previous work, we detected *PRKN* mRNA and Parkin protein in PBMCs from healthy controls [25, 26]. Parkin is a multi-functional protein with roles in diverse cellular processes, including those related to immunity [27]. Its expression in these cells is thus expected. However, we were unable to detect PINK1 protein in PBMCs, and levels of *PINK1* mRNA in these cells were remarkably low, up to 3000 times lower than those detected in fibroblast cells isolated from healthy controls. The low levels of PINK1 expression in PBMCs likely

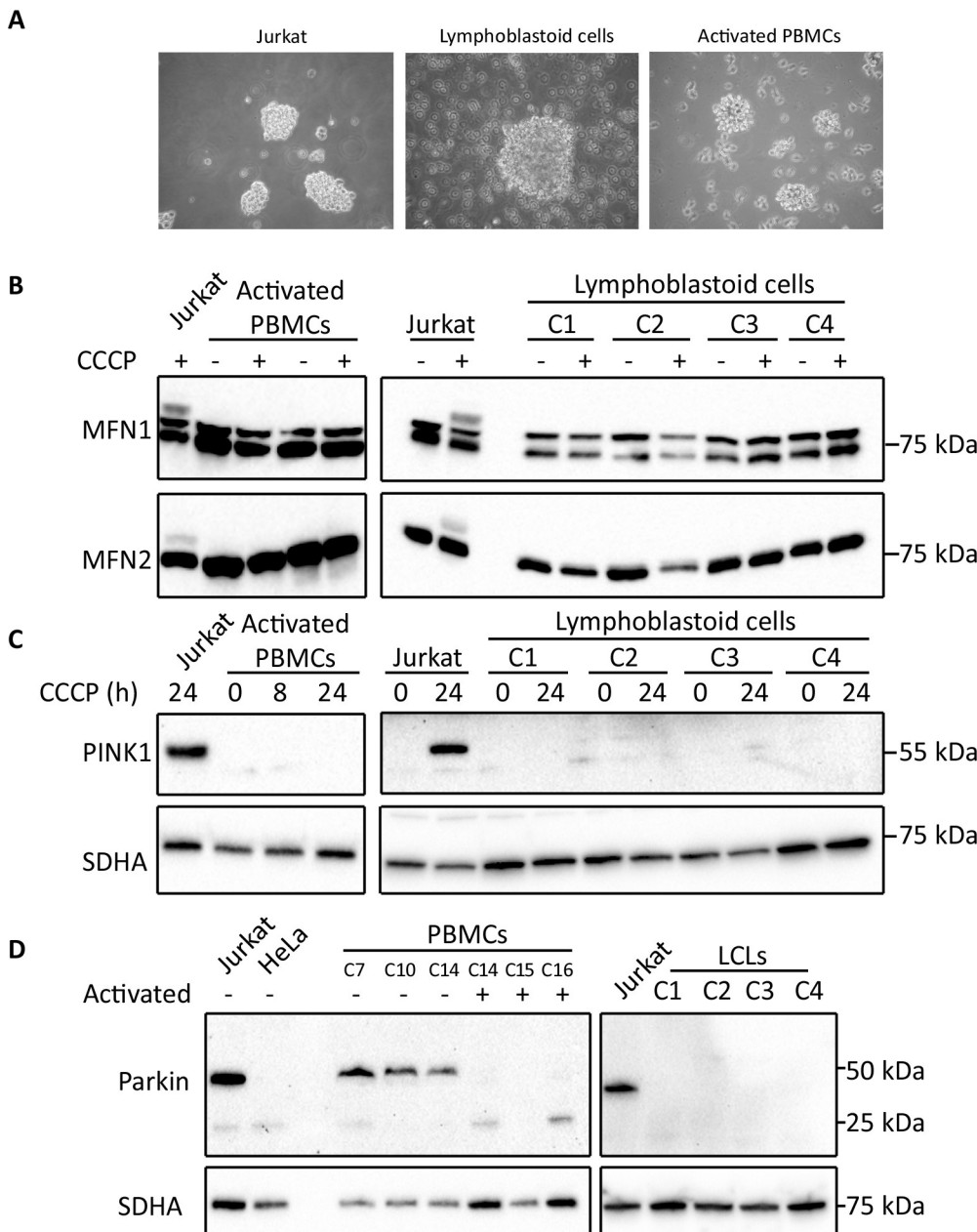

**Fig 8. Analysis of PINK1 –Parkin signalling pathway in lymphoblastoid cell lines and CD3/CD28 activated PBMCs.** (A) Phase contrast microscopy images of Jurkat cells, a lymphoblastoid cell line (LCL) and CD3/CD28 Dynabead-activated PBMCs. (B) Western blot analysis of MFN1 and -2 ubiquitination following CCCP treatment of activated PBMCs and LCLs. (C) Western blot analysis of PINK1 expression in activated PBMCs and LCLs following prolonged CCCP treatment. (D) Western blot analysis of Parkin expression in resting and CD3/CD28 Dynabead-activated PBMCs, and LCLs.

explains their lack of CCCP-induced MFN ubiquitination, as PINK1 accumulation is required to activate Parkin on the mitochondrial surface [8]. Indeed, CCCP treatment of PBMCs did not result in Parkin co-localisation to depolarised mitochondria. Moreover, PINK1 protein was undetectable in LCLs and mRNA levels were 10–25 times lower than those detected in fibroblasts.

Jurkat cells underwent CCCP-induced MFN ubiquitination and time dependent reduction in SDHA levels, indicating functional mitophagy similar to that observed in fibroblasts and SH-SY5Y cells. Moreover, Jurkat cells expressed readily detectable levels of *PINK1* and *PRKN* mRNA and the respective proteins. Jurkat cells are an EBV-negative T-cell line derived from an acute lymphoblastic leukaemia biopsy, and proliferate rapidly in culture [28, 29]. We thus considered that it may have been the quiescence of primary PBMCs which rendered them insensitive to CCCP-induced mitophagy. However, stimulation of PBMCs with CD3/CD28 antibodies, whilst leading to robust proliferation [30], did not result in CCCP-induced MFN ubiquitination. In fact, stimulated PBMCs massively downregulated Parkin expression. PBMCs isolated from patients are a heterogeneous cellular sample [31], and this reduction in Parkin expression post-stimulation may have been due to the clonal expansion of PBMC sub-populations lacking Parkin expression. In favour of this hypothesis, we noted that EBV-transformed lymphocytes from four independent controls did not express Parkin protein at detectable levels. It is tempting to speculate that the lack of Parkin in these immunoactive cells may reflect a specific function. For instance, the downregulation of Parkin in blood cells may promote viral clearance through promoting mitochondrial ROS production [32].

There are multiple pathways orchestrating mitophagy of mammalian mitochondria [33]. Some involve mitophagy receptors expressed on the OMM that directly interact with the autophagy machinery. Others function indirectly. Of these, the ubiquitin-dependent PINK1 – Parkin pathway is the best studied. The fact that *PINK1* and *PRKN* mutations are causative of early-onset PD underscores the importance of the PINK1 –Parkin mitophagy pathway in the brain but it has been demonstrated that PINK1 is dispensable for basal mitophagy [34, 35]. Our observations suggest that PBMCs employ PINK1-independent mitophagy pathways to clear defective mitochondria.

## Conclusions

Overall, our findings demonstrate PBMCs from control subjects do not undergo MFN ubiquitination after acute induction of mitophagy with CCCP. PBMCs express extraordinarily low *PINK1* mRNA levels and PINK1 protein is undetectable. The physiology of these cells thus precludes their use as a platform in studying the PINK1 –Parkin mitophagy signalling pathway in PD research.

## Methods

### Attainment of samples

Primary human dermal fibroblast cultures were established from skin explants of two early-onset PD patients, one female patient with a homozygous p.R246X nonsense mutation in *PINK1* and a heterozygous deletion of exons 4─6 in *PRKN*, and one female patient with a homozygous deletion of exons 4─5 in *PRKN*. In addition, skin biopsies were obtained from age-matched healthy control subjects. Fibroblast cultures were established according to standard procedures [36]. To generate Epstein-Barr virus (EBV) immortalised lymphoblastoid cell lines (LCLs), blood samples from healthy control subjects were sent to the European Collection of Authenticated Cell Cultures (ECACC) for transformation. Ethical approval for this work was obtained from the Royal Free Hospital and Medical School Research Ethics Committee (REC 07/H0720/161). Ethical approval for PBMC work was obtained from Camden and King's Cross Research Ethics Committee (REC 17/LO/1166). All donors provided prior informed written consent and all work was performed in compliance with national legislation and the Declaration of Helsinki.

SH-SY5Y neuroblastoma and Jurkat T-lymphocyte cell lines were purchased from the ECACC general cell collection (ECACC 94030304; ECACC 88042803).

## PBMC isolation, culture and treatment

PBMCs were isolated using standard Ficoll gradient separation protocols. Blood collected in EDTA-coated vacutainer blood tubes was mixed with an equal volume of phosphate-buffered saline (PBS). Diluted blood (20 ml) was layered on to 15 ml of Lymphoprep (Stemcell Technologies) in 50-ml Falcon tubes and centrifuged at $400\times g$ for 30 min without deceleration. The resultant PBMC layer was then aspirated into a new 15-ml Falcon tube and washed twice in PBS, centrifuging at $300\times g$ to reduce platelet contamination. Fresh PBMCs were then re-suspended in RPMI medium (Thermo Fisher Scientific, 61870) and counted using a flow cytometer (Moxi GO™, Orflo Technologies).

For routine analysis, PBMCs were cultured at $\sim 1\times10^6$ cells/ml in RPMI medium 1640 (Thermo Fisher Scientific, 61870–127), supplemented with 10% foetal bovine serum (FBS) and 1 mM sodium pyruvate at 37˚C, 5% $CO_2$. After 24 h, cells were treated with 20 μM CCCP (Sigma-Aldrich) and harvested at the indicated time points. For harvesting, 2 volumes of PBS were directly added to the suspended cells, followed by centrifugation at $500\times g$ for 10 min. The pellet was resuspended in 1 ml of PBS and centrifuged at $17,000\times g$ for 10 min at 4˚C. The supernatant was aspirated and the pellet was stored at -80˚C until further analysis. The age and sex of the donors of the PBMC samples used in this study are shown in S1 Table.

## Culturing of fibroblasts, SH-SY5Y, Jurkat and LCLs

Primary human skin fibroblast cultures were grown in DMEM (Thermo Fisher Scientific, 61965–059), supplemented with 10% FBS, 1 mM sodium pyruvate, and 50 units/ml of penicillin and 50 μg/ml of streptomycin at 37˚C, 5% $CO_2$. SH-SY5Y cells were cultured in DMEM: F12 (1:1) (Thermo Fisher Scientific, 31331–028), supplemented with 10% FBS, 1× non-essential amino acids and penicillin/streptomycin at 37˚C, 5% $CO_2$. When confluent, cultures were passaged using trypsinisation. Jurkat cells and LCLs were cultured in suspension in RPMI medium 1640 (Thermo Fisher Scientific, 61870–127), supplemented with 10% FBS and 1 mM sodium pyruvate at 37˚C, 5% $CO_2$. Cells were maintained at $5\times10^5$–$3\times10^6$ cells/ml.

## PBMC activation with CD3/CD28 Dynabeads

PBMCs were incubated with Dynabeads Human T-Activator CD3/CD28 (Thermo Fisher Scientific, 11132D) at a ratio of 1:1 and cultured in OpTmizer T-cell expansion SFM medium, supplemented with 10 mM L-glutamine, penicillin/streptomycin, serum substitute (Thermo Fisher Scientific, A1048501) and 30 U/ml of interleukin-2 (IL-2; Thermo Fisher Scientific, PHC0026) at 37˚C, 5% $CO_2$. Cultures were maintained at $1\times10^6$—$3\times10^6$ cells/ml. Cells were activated for 7 days prior to analysis.

## Reverse-transcriptase real-time quantitative PCR

RNA was isolated from cell pellets with the RNeasy kit (Qiagen, 74104) and quantified with a Nanodrop spectrophotometer (Thermo Fisher Scientific). RNA (500 ng) was reverse transcribed to produce cDNA libraries using the QuantiTect Reverse Transcriptase Kit from Qiagen (205311). cDNA was quantified by Qubit analysis with the 1× dsDNA HS Assay Kit (Thermo Fisher Scientific, Q33230). For qPCR, 1 ng of cDNA was mixed with forward and reverse primers (final concentration 10 μM), PowerUp SYBR Green Master Mix (Thermo

Fisher Scientific, A25780) and water to a final volume of 20 μl. Primers were purchased from Qiagen (*PRKN*: QT00023401; *PINK1*: QT00056630).

Following initial denaturation at 50˚C for 2 min and activation at 95˚C for 2 min, thermo-cycling was performed with 50 cycles of denaturation (94˚C, 15 sec), annealing (55˚C, 30 sec) and extension (72˚C, 30 sec), using an Applied Biosystems StepOne real-time PCR system. Samples were analysed as technical triplicates. Fluorescence was read during the extension step. Melting temperature analysis was performed on the amplified products to ensure consistency within and between runs. Amplicon specific $C_t$ thresholds were applied consistently between runs, and relative transcript levels were calculated by transforming the $C_t$ values using the expression $2^{-Ct}$. As we compared transcript levels in different cell types, we could not use transcript levels of household genes, such as *GAPDH* or *ACTB*, for normalisation, because household genes are transcribed at variable levels in different cell types. Therefore, we normalised $C_t$ values to cDNA quantity after accurate Qubit fluorometric quantitation.

## Cell lysis and western blotting

Cell pellets were lysed in 0.1% Triton X-100 in PBS containing a Halt Protease Inhibitor Cocktail (Thermo Fisher Scientific; 78430). Lysates were vortexed and incubated for 15 min on ice, followed by clarification at 17,000× *g* for ten min at 4˚C. Following determination of the protein concentration in the supernatants (Pierce™ BCA Protein Assay Kit; Thermo Fisher Scientific, 23250), samples were prepared for denaturing gel electrophoresis by addition of Laemmli Sample Buffer (BioRad, 161–0747), Sample Reducing Agent (Thermo Fisher Scientific, NP0009) and water to consistent protein concentrations.

Samples were resolved on Mini-Protean® TGX 4–20% gels or TGX 7.5% gels (BioRad Laboratories, 4568095 and 4568025). Separated proteins were transferred to Trans-Blot Turbo 0.2-μm PVDF membranes (BioRad, 170–4157), using the BioRad Trans-Blot Turbo Transfer System. Membranes were blocked in 10% non-fat dry milk powder in PBS prior to incubation with primary antibodies overnight at 4˚C in 5% milk, 0.15% Tween-20, PBS. Following washing with 0.15% Tween-20, PBS, membranes were incubated with secondary antibodies conjugated to HRP enzymes (Dako, P0447 and P0448) and washed again. Blots were developed with Clarity Western ECL Substrate (BioRad, 170–5060). Capturing of the chemiluminescent signals was performed with the BioRad Chemidoc™ MP Imaging System. Signals were quantified with BioRad Image Lab 6.0.1 software. The primary antibodies are specified in S2 Table.

## Visualisation of polarised mitochondria with TMRM

Jurkat cells and PBMC samples from donors C17 and C18 (S1 Table) were seeded in 35-mm μ-dishes with a glass bottom (Ibidi) with phenol red free RPMI medium 1640 (Thermo Fisher Scientific, 32404–014), supplemented with 1 mM GlutaMax (Thermo Fisher Scientific), 10% FBS and 1 mM sodium pyruvate. TMRM (Thermo Fisher Scientific) was added to the cell suspensions to a final concentration of 25 nM to stain polarised mitochondria fluorescent red. Hoechst 33343 (Thermo Fisher Scientific) was added to a final concentration of 2 μM to counterstain nuclei fluorescent blue. The fluorescent signals were recorded with a Nikon Eclipse Ti-E inverted confocal laser-scanning microscope, equipped with a ×60 objective. After 15 min, CCCP to a final concentration of 20 μM was added, whilst recording was continued for a further 3 min. Imaging data were collected with NIS-Elements software (Nikon). To construct the figures, 7 z-stacks of 0.1 μm were projected with Image J software (National Institutes of Health).

## Immunocytochemical localisation of Parkin

Jurkat cells and PBMC samples from donors C17 and C18 (S1 Table) were cultured in RPMI medium 1640 (Thermo Fisher Scientific, 61870–127), supplemented with 10% FBS and 1 mM sodium pyruvate at 37˚C, 5% $CO_2$. Dimethyl sulphoxide (DMSO, vehicle) or CCCP were added to a final concentration of 0.1% (v/v) or 20 μM, respectively, and cells were cultured for a further 50 min. Cells were collected by 10-min centrifugation at 500× $g$ and resuspended in PBS containing either 0.1% (v/v) DMSO or 20 μM CCCP. Cells were adhered to glass cover-slips by gravity sedimentation [37]. After 30 min, fluid was carefully aspirated and adhered cells were fixed with 4% paraformaldehyde in PBS for 20 min. Coverslips were washed in PBS, followed by 20-min antigen retrieval incubation in 10 mM sodium citrate buffer (pH 6.0) at 90˚C. After a wash in PBS, cells were permeabilised in methanol at -20˚C for 15 min, followed by another wash in PBS. Samples were blocked with 10% (v/v) normal goat serum in PBS in a humidified atmosphere at 37˚C for 30 min. Next, samples were incubated with anti-Parkin and anti-TOMM20 primary antibodies (S2 Table) in a humidified atmosphere at 37˚C for 45 min. After further washes with PBS, samples were incubated with secondary antibodies conjugated to Alexa Fluor 488 or Alexa Fluor 594 (Thermo Fisher Scientific, A32723 and A32740) and washed again. Finally, coverslips were mounted onto glass slides in Citifluor AF2 mounting medium (Agar Scientific), containing 1 μg/ml of 4′,6-diamidino-2-phenylindole (DAPI, Sigma-Aldrich) to counterstain nuclei blue. Slides were examined with a Nikon Eclipse Ti-E inverted confocal laser-scanning microscope, equipped with a ×60 objective. Images (single z-stacks of 0.1 μm) were captured with NIS-Elements software (Nikon). Image J software was used to construct the figures.

## Statistical analyses

Graphs and statistical analyses were executed with GraphPad Prism® version 6.01 software. Data are presented as mean ± standard error of the mean. Student's t test was used to examine statistical significance and statistical significance levels were set to $p < 0.05$.

## Supporting information

**S1 Fig. Time course analysis of MFN ubiquitination in Jurkat cells.** Jurkat cells were treated with 20 μM CCCP for increasing periods of time and the ubiquitination of MFN1/MFN2 was analysed by immunoblotting. The levels of ubiquitinated MFN1 and MFN2 were quantified relative to the respective non-ubiquitinated protein. Graph represents mean ± SEM; MFN1, n = 2; MFN2, n = 3.
(PDF)

**S2 Fig. Analysis of PINK1 expression in different cell types.** SH-SY5Y, fibroblast, Jurkat and PBMC cultures were left untreated or treated with 20 μM CCCP for 24 hours. Samples were analysed by western blotting with the PINK1 antibody clone D8G3 (Cell Signaling Technology). PINK1 protein is indicated by the red box.
(PDF)

**S3 Fig. Reverse transcriptase qPCR analysis of *PINK1* in LCLs.** qPCR was performed on equal amounts of cDNA from each cell sample, and $C_t$ values were expressed relative to the fibroblast (Fib) sample.
(PDF)

**S1 Table. Age and sex of PBMC donors.**
(PDF)

**S2 Table. Characteristics of primary antibodies used in this study.**
(PDF)

**S1 File. Calculations for Figs 5A and 5D, and 6A, 6C and 6D.**
(XLSX)

**S1 Raw images. Uncropped, unadjusted images of western blots.**
(PDF)

# Acknowledgments

We would like to thank Dr Bledi Petriti for some of the PMBC preparation. We would also like to acknowledge the Research Assistants John Harvey and Miriam Pollard for their work on the wider project of which this study was a part.

# Author Contributions

**Conceptualization:** Jan-Willem Taanman.

**Data curation:** Jan-Willem Taanman.

**Formal analysis:** Aaron V. Bradshaw, Jan-Willem Taanman.

**Funding acquisition:** Anthony H. V. Schapira, Jan-Willem Taanman.

**Investigation:** Aaron V. Bradshaw, Jan-Willem Taanman.

**Methodology:** Aaron V. Bradshaw, Philip Campbell, Jan-Willem Taanman.

**Project administration:** Jan-Willem Taanman.

**Resources:** Philip Campbell, Anthony H. V. Schapira, Huw R. Morris.

**Supervision:** Jan-Willem Taanman.

**Validation:** Aaron V. Bradshaw.

**Visualization:** Aaron V. Bradshaw, Jan-Willem Taanman.

**Writing – original draft:** Aaron V. Bradshaw.

**Writing – review & editing:** Philip Campbell, Anthony H. V. Schapira, Huw R. Morris, Jan-Willem Taanman.

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
