## [Decision Letter · Decision Letter 0]

22 Oct 2021

PONE-D-21-29921The PINK1 – Parkin mitophagy signalling pathway is not functional in peripheral blood mononuclear celPLOS ONE

Dear Dr. Taanman,

Thank you for submitting your manuscript to PLOS ONE. After careful consideration, we feel that it has merit but does not fully meet PLOS ONE’s publication criteria as it currently stands. Therefore, we invite you to submit a revised version of the manuscript that addresses the points raised during the review process.

With respect to Reviewer 2 comments:

1. if Ub blots are available I strongly recommend that they be included to strengthen the argument that the upper Mfn band is ubiquitinated.

2. I agree that while your data do show that Pink-Parkin signaling is not active in PBMCs, your results more specifically show that PINK1 is absent. This is in and of itself interesting as Pink1 mediates mitophagy by other Ub ligases other than Parkin. While I recognize that the focus of the paper is to show that PBMC screening would not be appropriate for PD, I would recommend that the manuscript mention the existence of Pink/Parkin independent mitophagy pathways. Indeed, a number of studies have shown that Pink is dispensable for basal mitophagy

- McWilliams TG, Prescott AR, Montava-Garriga L, Ball G, Singh F, Barini E, et al. Basal Mitophagy Occurs Independently of PINK1 in Mouse Tissues of High Metabolic Demand. Cell Metab 2018; 27:439-49 e5.

- Lee JJ, Sanchez-Martinez A, Zarate AM, Beninca C, Mayor U, Clague MJ, et al. Basal mitophagy is widespread in Drosophila but minimally affected by loss of Pink1 or parkin. J Cell Biol 2018; 217:1613-22.

3. an interesting suggestion but not necessary.

We look forward to receiving your revised manuscript.

Kind regards,

Ivan R. Nabi, Ph.D.

Academic Editor

PLOS ONE

Journal Requirements:

 [JWT, 15271, Michael J. Fox Foundation for Parkinson’s Research, https://www.michaeljfox.org/, NO

JWT, 42, Royal Free Charity, https://royalfreecharity.org/, NO

AHVS, 668738, European Union Horizon 2020 Research and Innovation programme https://ec.europa.eu/programmes/horizon2020/en/home, NO

AHVS, MR/M006646/1, Medical Research Council, https://mrc.ukri.org/, NO].  

[I have read the journal's policy and the authors of this manuscript have the following competing interests: JWT received funding from Michael J. Fox Foundation for Parkinson’s Research and the

Royal Free Charity. AHVS received funding from the European Union and the Medical Research Council, and is a Consultant for Sanofi Aventis. HM is a Consultant for AlzProtect, Accorda, Bristol-Myers-Squibb, E-scape and the Wellcome Trust, and has received lecture fees from GE Healthcare, GSK, UCB Pharma, and the Wellcome Trust.] 

Reviewers' comments:

Reviewer's Responses to Questions

**Comments to the Author**

1. Is the manuscript technically sound, and do the data support the conclusions?

Reviewer #1: Yes

Reviewer #2: Partly

2. Has the statistical analysis been performed appropriately and rigorously? 

Reviewer #1: Yes

Reviewer #2: Yes

3. Have the authors made all data underlying the findings in their manuscript fully available?

Reviewer #1: Yes

Reviewer #2: Yes

4. Is the manuscript presented in an intelligible fashion and written in standard English?

Reviewer #1: Yes

Reviewer #2: Yes

5. Review Comments to the Author

Reviewer #1: In this revision, the authors have responded to both reviewers comments very well and the manuscript is much improved. They have added additional data where required which has more strongly supported their conclusions.

Reviewer #2: The work on "The PINK1 – Parkin mitophagy signalling pathway is not functional in peripheral blood mononuclear cells" by Jan-Willem Taanman et al is very interesting.

Here I have some suggestions:

(1) The detection of ubiquitination of mitofusins can be more convincing if anti-ubiquitin WB is shown side by side

(2)The statement "The PINK1 – Parkin mitophagy signalling pathway is not functional in peripheral blood mononuclear cells" is too strong. The work shows the PINK1 – Parkin mitophagy signalling pathway is very limited due to the low level of PINK1

(3)Different cells may have different level responses to the stress (4 hours to 24 hours) or different treatments (CCCP, antimycin A, valinomycin, and rotenone). Have the authors tested them?

6. PLOS authors have the option to publish the peer review history of their article (what does this mean?). If published, this will include your full peer review and any attached files.

Reviewer #1: No

Reviewer #2: No

---

## [Author Response · Author response to Decision Letter 0]

25 Oct 2021

REBUTTAL

PONE-D-21-29921

The PINK1 – Parkin mitophagy signalling pathway is not functional in peripheral blood mononuclear cells

PLOS ONE

Dear Dr. Taanman,

Thank you for submitting your manuscript to PLOS ONE. After careful consideration, we feel that it has merit but does not fully meet PLOS ONE’s publication criteria as it currently stands. Therefore, we invite you to submit a revised version of the manuscript that addresses the points raised during the review process.

With respect to Reviewer 2 comments:

1. if Ub blots are available I strongly recommend that they be included to strengthen the argument that the upper Mfn band is ubiquitinated.

We agree with Reviewer #2 that it is important to demonstrate that the slower migrating bands recognised by anti-MFN1 and anti-MFN2 antibodies in CCCP-treated samples represent ubiquitinated mitofusin. We were the first to demonstrate that MFN1 and MFN2 are ubiquitinated in in a PINK1/Parkin-dependent manner upon induction of mitophagy [Gegg et al. Hum Mol Genet. 2010;19:4861-70; Ref. 10]. In our 2010 paper, we demonstrated that these slower migrating bands in CCCP-treated cells represent ubiquitinated mitofusin by means of anti-mitofusin pull-down immunoprecipitation experiments with an anti-ubiquitin antibody (Fig 4C and D in Gegg et al., 2010). The current manuscript builds on our previous observations. We have indicated this in the revised manuscript more clearly (line 102─104): “This size change is consistent with mono-ubiquitination and our previous work anti-mitofusin immunoprecipitation experiments [10] indicated that the extra MFN1 and MFN2 bands detected post-CCCP treatment are ubiquitin positive”.

2. I agree that while your data do show that Pink-Parkin signaling is not active in PBMCs, your results more specifically show that PINK1 is absent. This is in and of itself interesting as Pink1 mediates mitophagy by other Ub ligases other than Parkin. While I recognize that the focus of the paper is to show that PBMC screening would not be appropriate for PD, I would recommend that the manuscript mention the existence of Pink/Parkin independent mitophagy pathways. Indeed, a number of studies have shown that Pink is dispensable for basal mitophagy

- McWilliams TG, Prescott AR, Montava-Garriga L, Ball G, Singh F, Barini E, et al. Basal Mitophagy Occurs Independently of PINK1 in Mouse Tissues of High Metabolic Demand. Cell Metab 2018; 27:439-49 e5.

- Lee JJ, Sanchez-Martinez A, Zarate AM, Beninca C, Mayor U, Clague MJ, et al. Basal mitophagy is widespread in Drosophila but minimally affected by loss of Pink1 or parkin. J Cell Biol 2018; 217:1613-22.

We have included a paragraph mentioning the existence of Pink/Parkin independent mitophagy pathways in the revised manuscript (lines 328─334): “There are multiple pathways orchestrating mitophagy of mammalian mitochondria [33]. Some involve mitophagy receptors expressed on the OMM that directly interact with the autophagy machinery. Others function indirectly. Of these, the ubiquitin-dependent PINK1 – Parkin pathway is the best studied. The fact that PINK1 and PRKN mutations are causative of early-onset PD underscores the importance of the PINK1 – Parkin mitophagy pathway in the brain but it has been demonstrated that PINK1 is dispensable for basal mitophagy [34, 35]. Our observations suggest that PBMCs employ PINK1-independent mitophagy pathways to clear defective mitochondria”.

3. an interesting suggestion but not necessary.

See below.

We look forward to receiving your revised manuscript.

Kind regards,

Ivan R. Nabi, Ph.D.

Academic Editor

PLOS ONE

Comments to the Author

1. Is the manuscript technically sound, and do the data support the conclusions?

Reviewer #1: Yes

Reviewer #2: Partly

2. Has the statistical analysis been performed appropriately and rigorously? 

Reviewer #1: Yes

Reviewer #2: Yes

3. Have the authors made all data underlying the findings in their manuscript fully available?

Reviewer #1: Yes

Reviewer #2: Yes

4. Is the manuscript presented in an intelligible fashion and written in standard English?

Reviewer #1: Yes

Reviewer #2: Yes

5. Review Comments to the Author

Reviewer #1: In this revision, the authors have responded to both reviewers comments very well and the manuscript is much improved. They have added additional data where required which has more strongly supported their conclusions.

We thank Reviewer #1 for her/his kind words.

Reviewer #2: The work on "The PINK1 – Parkin mitophagy signalling pathway is not functional in peripheral blood mononuclear cells" by Jan-Willem Taanman et al is very interesting.

Here I have some suggestions:

(1) The detection of ubiquitination of mitofusins can be more convincing if anti-ubiquitin WB is shown side by side

See above.

(2) The statement "The PINK1 – Parkin mitophagy signalling pathway is not functional in peripheral blood mononuclear cells" is too strong. The work shows the PINK1 – Parkin mitophagy signalling pathway is very limited due to the low level of PINK1

See above.

(3) Different cells may have different level responses to the stress (4 hours to 24 hours) or different treatments (CCCP, antimycin A, valinomycin, and rotenone). Have the authors tested them?

Our standard treatment to induce acute mitophagy in PBMCs was 2 h, 20 �M CCCP. However, we have tested a range of CCCP treatment times (0─24 h) and CCCP concentrations (0─100 �M). In addition, we have stressed PBMCs prior and during CCCP treatment by culturing for up to 8 h in Hank’s balanced salt solution or by culturing for up to 7 d in medium is which glucose was substituted by gala

---

## [Editor Report · Decision Letter 1]

29 Oct 2021

The PINK1 – Parkin mitophagy signalling pathway is not functional in peripheral blood mononuclear cells

PONE-D-21-29921R1

Dear Dr. Taanman,

We’re pleased to inform you that your manuscript has been judged scientifically suitable for publication and will be formally accepted for publication once it meets all outstanding technical requirements.

Kind regards,

Ivan R. Nabi, Ph.D.

Academic Editor

PLOS ONE
---

## [Editor Report · Acceptance letter]

2 Nov 2021

PONE-D-21-29921R1 

The PINK1 – Parkin mitophagy signalling pathway is not functional in peripheral blood mononuclear cells 

Dear Dr. Taanman:

I'm pleased to inform you that your manuscript has been deemed suitable for publication in PLOS ONE. Congratulations! Your manuscript is now with our production department. 

Kind regards, 

on behalf of

Dr. Ivan R. Nabi 

Academic Editor

PLOS ONE